# Lidar- and UAV-Based Vertical Observation of Spring Ozone and Particulate Matter in Nanjing, China

Yawei Qu [1,2,†], Ming Zhao [3,†], Tijian Wang [4,*], Shu Li [4], Mengmeng Li [4], Min Xie [4] and Bingliang Zhuang [4]

1. College of Intelligent Science and Control Engineering, Jinling Institute of Technology, Nanjing 211169, China; yawei.qu@jit.edu.cn
2. Key Laboratory of Meteorological Disaster (KLME), Ministry of Education & Collaborative Innovation Center on Forecast and Evaluation of Meteorological Disasters (CIC-FEMD), Nanjing University of Information Science & Technology, Nanjing 210044, China
3. Key Laboratory of Atmospheric Optics, Anhui Institute of Optics and Fine Mechanics, Hefei Institutes of Physical Science, Chinese Academy of Sciences, Hefei 230031, China; zhaom@aiofm.ac.cn
4. School of Atmospheric Sciences, Nanjing University, Nanjing 210023, China; lishu@nju.edu.cn (S.L.); mengmengli2015@nju.edu.cn (M.L.); minxie@nju.edu.cn (M.X.); blzhuang@nju.edu.cn (B.Z.)
* Correspondence: tjwang@nju.edu.cn
† These authors contributed equally to this work.

**Abstract:** The rapid urbanization in China is accompanied by increasingly serious air pollution. Particulate matter and ozone are the main air pollutants, and the study of their vertical distribution and correlation plays an important role in the synergistic air pollution control. In this study, we performed Lidar- and UAV-based observations in spring in Nanjing, China. The average concentrations of surface ozone and $PM_{2.5}$ during the observation period are 87.78 $\mu g \, m^{-3}$ and 43.48 $\mu g \, m^{-3}$, respectively. Vertically, ozone reaches a maximum in the upper boundary layer, while the aerosol extinction coefficient decreases with height. Generally, ozone and aerosol are negatively correlated below 650 m. The correlation coefficient increases with altitude and reaches a maximum of 0.379 at 1875 m. Within the boundary layer, ozone and aerosols are negatively correlated on days with particulate pollution ($PM_{2.5} > 35 \, \mu g \, m^{-3}$), while on clean days they are positively correlated. Above the boundary layer, the correlation coefficient is usually positive, regardless of the presence of particulate pollution. The UAV study compensates for Lidar detections below 500 m. We found that ozone concentration is higher in the upper layers than in the near-surface layers, and that ozone depletion is faster in the near-surface layers after sunset.

**Keywords:** particulate matter; ozone; Lidar; UAV

## 1. Introduction

Fine particulate matter ($PM_{2.5}$) is an important atmospheric pollutant that has important effects on human health. Short-term or long-term exposure to particulate matter pollution can lead to respiratory diseases, cardiovascular diseases, lung cancer, and even death [1]. Tropospheric ozone ($O_3$) is also an important atmospheric pollutant. As one of the major greenhouse gases, it can harm human health, crop growth, and impact regional climate [2]. Tropospheric ozone, also named 'bad ozone', is mainly from stratospheric transport, and photochemical production [3]. In recent years, as a result of the implementation of emission control policies in China, $PM_{2.5}$ pollution has been effectively mitigated [4,5], however, ozone pollution has become a severe environmental problem, especially in major urban agglomerations, such as Yangtze River Delta (YRD) and Beijing–Tianjin–Hebei (BTH) [6,7].

Rather than two isolated air pollution issues, $PM_{2.5}$ and ozone pollution are complexly correlated. Most ozone precursors come from the combustion of fossil fuels and biomass fuels, which have some similarity to the sources of $PM_{2.5}$. Moreover, there are multiple

interactions between $PM_{2.5}$ and ozone, that is, $PM_{2.5}$ affects ozone concentration by influencing photolysis rates [8], heterogeneous reactions [9], and planet boundary layer (PBL) development [10,11], while $O_3$ can change the oxidative capacity of the atmosphere and affect the production of secondary $PM_{2.5}$ [12]. In order to solve the current $PM_{2.5}$ and $O_3$ combined air pollution, the first and most fundamental step is to fully understand the distribution and correlation of $PM_{2.5}$ and $O_3$ in the urban boundary layer. Previous studies have applied various methods to detect the vertical distribution of the two pollutants.

Site observation and satellite data retrieval are widely used for air pollution monitoring. Site observation is the most traditional method, and in the vertical direction, meteorological tower and sounding balloons are usually used [13–15]. However, the height of the meteorological tower and the number of observation sites are limited. The frequency of balloon release is limited and highly influenced by the weather condition, which makes it more difficult to obtain continuous vertical profiles of pollutants. With the launch of Earth-observing satellites, satellite data such as OMI, MODIS, and CALIPSO have been widely used to study the vertical distribution of pollutants. Zhou et al. [16] retrieved the vertical profile of aerosol extinction coefficient in East Asia using CALIPSO data and found that the extinction coefficient decreased exponentially with height under high particle concentration conditions and increased exponentially under low particle concentration conditions. Shen et al. [17] used OMI data to analyze boundary layer ozone in China and found elevated mean ozone concentrations in eastern China from 2013 to 2017. However, satellite observations were sensitive to clouds and upper-layer aerosols, and the vertical resolution was relatively low within the troposphere, especially in the PBL [18].

In recent years, new remote sensing techniques such as ground-based Lidar and UAVs have been applied to air pollution detection. Tao et al. [19] detected the aerosol extinction coefficient based on CCD side-scattering Lidar and found that the vertical distribution of $PM_{2.5}$ within the PBL was complex with a multilayer structure, and when $PM_{2.5}$ pollution was heavy, the particulate matter was lifted to higher altitudes and transported to other regions. By performing a ground-based ozone Lidar observation in Shanghai, Xing et al. [20] found that ozone was not only generated at the surface, but also strongly generated at an altitude of about 1100 m, and the local ozone was mainly affected by the abundance of volatile organic compounds (VOCs) in the lower troposphere in East China. In previous studies, Lidar $PM_{2.5}$ concentration inversion usually used a traditional linear model, the accuracy of which is relatively low. Ma et al. [21] has found that machine learning algorithms can improve the accuracy of Lidar measurement inversion and benefit the application of Lidar in air quality monitoring. Ground-based Lidar detection does not have the problem of near-surface detection accuracy in satellite observation and the detection frequency problem in balloon sounding, but there is still limitation in Lidar detection within a few hundred meters near the ground due to overlap.

With the development of Unmanned Aerial Vehicle (UAV) technology and sensor integration, UAV-based atmospheric environment monitoring has been gradually used. UAV-based detection can compensate for the overlap area of Lidar [22], allowing for more complete vertical profiles of $PM_{2.5}$ and ozone. Chang et al. [23] collected aerial samples at 300 m using a six-rotor UAV and compared them with surface air samples, and found large differences in pollutant concentration and composition between the surface and aerial samples, implying an inhomogeneous mixing, under local circulation and high-pressure peripheral circulation conditions. Wu et al. [24] used a multi-rotor UAV to detect the vertical distribution of black carbon aerosol (BC) and ozone concentrations in Shenzhen and found that BC concentration decreased with height, while ozone concentration increased, and the vertical profiles of BC and ozone were affected by both planet boundary layer height (PBLH) and air mass source.

In previous studies, observational studies for $PM_{2.5}$ and ozone have made great progress. However, limited to the spatial or temporal scales, the completeness and accuracy of the pollutant concentration profiles obtained by a single observation method are limited. Therefore, in this study, we carried out multi-method boundary layer observations

including Lidar-, UAV-, and ground-based observations to reveal the vertical distribution characteristics and correlations of PM$_{2.5}$ and ozone in spring in Nanjing, eastern China. Section 2 presents the experiments, observational instrumentation, and data validation. Observation results and discussion are included in Sections 3 and 4, respectively. Finally, conclusions are given in Section 5.

## 2. Materials and Methods

### 2.1. Lidar System

The experiments were performed at Nanjing University Xianlin campus (32.12° N, 118.95° E), eastern suburbs of Nanjing, from 4 April to 27 May 2019. The vertical distributions of ozone and aerosol extinction coefficient were continuously measured by the LGO-01 ozone Lidar system produced by Anhui Lanke Information Technology Co., Ltd., Chuzhou, China. The system structure of ozone Lidar is shown in Figure 1. A solid-state Nd:YAG (neodymium-doped yttrium aluminum garnet) was utilized as a laser source and emitted simultaneously light pulses of 266 nm and 532 nm wavelength. The 266 nm laser was transmitted into a Raman cell filled with deuterium and produced two Raman shifted frequencies, the 289 nm and 316 nm laser beams. In the receiving unit, a Cassegrain telescope with a diameter of 300 mm was used to collect the backscattered signals. The aerosol extinction coefficient was derived from the backscattered signal of 532 nm wavelength through the Fernald method [25]. The planetary boundary layer height (PBLH) and the cloud bottom height were derived from the 532 nm signals using the wavelet algorithm [10]. Since ozone has different absorption cross-sections at 289 nm and 316 nm, the ozone concentration was derived from the backscattered signals at these two wavelengths. The raw spatial resolution was 7.5 m and the signal range was from 120 m to 3000 m. Due to the Lidar system's overlap, the data below 500 m were not used in this study. The detected signal for each measurement was the average of 3000 laser pulses, and the time span of each measurement cycle was set to 15 min. The system parameters are listed in Table 1.

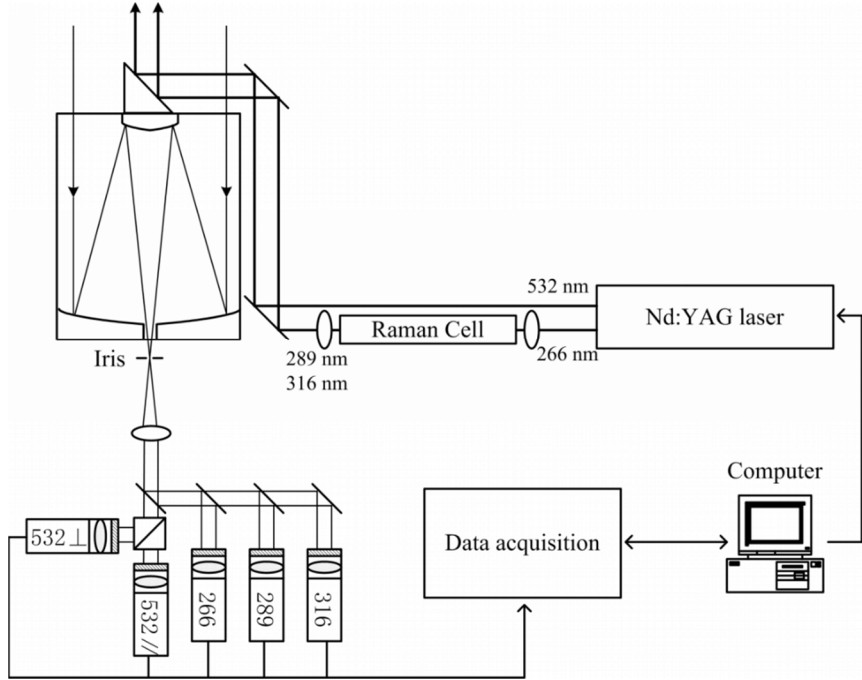

**Figure 1.** Schematic diagram of the Lidar system.

**Table 1.** Key parameters of the Lidar system.

| Parameters | Value |
| --- | --- |
| Transmitter | |
| Wavelength | 532/266/289/316 nm |
| Pulse energy | 150/100/10/8 mJ |
| Pulse frequency | 10 Hz |
| Pulse duration | <10 ns |
| Divergence | <0.5 mrad |
| Receiver | |
| Diameter | 300 mm |
| FOV | 0.5–2 mrad |
| Filter bandwidth | 1 nm |
| Data acquisition | |
| Sample rate | 20 MHz |
| Resolution | 12 bit |

### 2.2. UAV and Mobile Observation Vehicle

The M6E six-rotor UAV produced by Beijing Tiantong Aviation Technology Co., Ltd. (Beijing, China) was used in this study to detect near-surface atmospheric pollutant concentrations. The UAV is equipped with the RGK-3 multi-parameter measuring instrument produced by Qingdao Rongguang Electronic Technology Co., Ltd. (Qingdao, China), which can be used to measure air temperature, pressure, ozone concentration, and $PM_{2.5}$ and $PM_{10}$ mass concentration. The ozone sensor is the 7NE/O3-1 diffusion electrochemical sensor produced by Shengkaian Co. (Shenzhen, China) and the particulate matter sensor adopts the laser particle sensor produced by Temtop (Shenzhen, China). The observational data were updated every 63 s via wireless data transmission.

A mobile observation vehicle was parked near the Lidar system as a stationary site for surface observation. A continuous particulate monitor BPM-200, produced by Focused Photonics, Inc. (Hangzhou, China), was equipped to measure ambient $PM_{10}$ concentration by using the β-ray attenuation technique, and the accuracy was 0.1 μg m$^{-3}$. The measurement of $PM_{2.5}$ could also be achieved by introducing a corresponding cyclone. An ozone analyzer AQMS-300, produced by Focused Photonics, Inc., was equipped to measure ambient ozone concentration in ppb level by utilizing ultraviolet photometry. A sonic weather sensor SWS-600, produced by Hangzhou Pengpu Technology Co., Ltd. (Hangzhou, China) was used for integrated monitoring of wind speed, wind direction, rainfall, ambient temperature, relative humidity, and atmospheric pressure.

### 2.3. Data Verification

To verify the accuracy of Lidar-detected and UAV-based vertical profiles of pollutants, we compared the Lidar data with simultaneous observation results from UAV and the mobile observation vehicle.

The UAV-based experiment was carried out on 20 May 2019. The UAV, equipped with an ozone electrochemical sensor and a light-scattering particle counter, slowly ascended from the ground to an altitude of 1000 m from 18:01 to 18:23 LST. The vertical distribution of the ozone and particle extinction coefficients collected by the UAV during its ascent period were compared with the Lidar observations during the same period. Data collected during the UAV's landing were not included in the comparative experiments because the landing process was too fast for the sensor to respond. Figure 2a shows the ozone profile obtained by UAV (solid line) and Lidar (dashed line), where the UAV data were interpolated at 7.5 m intervals to match the spatial resolution of Lidar, and the Lidar profile was the average of four ozone profiles collected from 17:40 to 18:40 LST. Figure 2b shows the relative deviation of UAV and Lidar ozone profiles. The ozone profiles from the two methods were in good agreement from 600 m to 1000 m, with a relative deviation of less than 5%. However, due to the difference in the overlap of Lidar 289 nm and 316 nm signals, the Lidar data were inaccurate below 500 m, which was different from the UAV data, and the relative deviation

reached a maximum of 23.9%. Therefore, the Lidar ozone data below 500 m were not used in the following discussion. Figure 2c shows the $PM_{2.5}$ profile obtained by UAV (solid line) and Lidar (dashed line), where the Lidar $PM_{2.5}$ data were derived from the aerosol extinction coefficient, and Figure 2d shows the relative deviation of the two $PM_{2.5}$ profiles. The $PM_{2.5}$ concentration profiles observed above 500 m by UAV and Lidar were similar, with a relative deviation of less than 20%. Due to the geometric overlap of Lidar, $PM_{2.5}$ values below 500 m were inaccurate and were discarded.

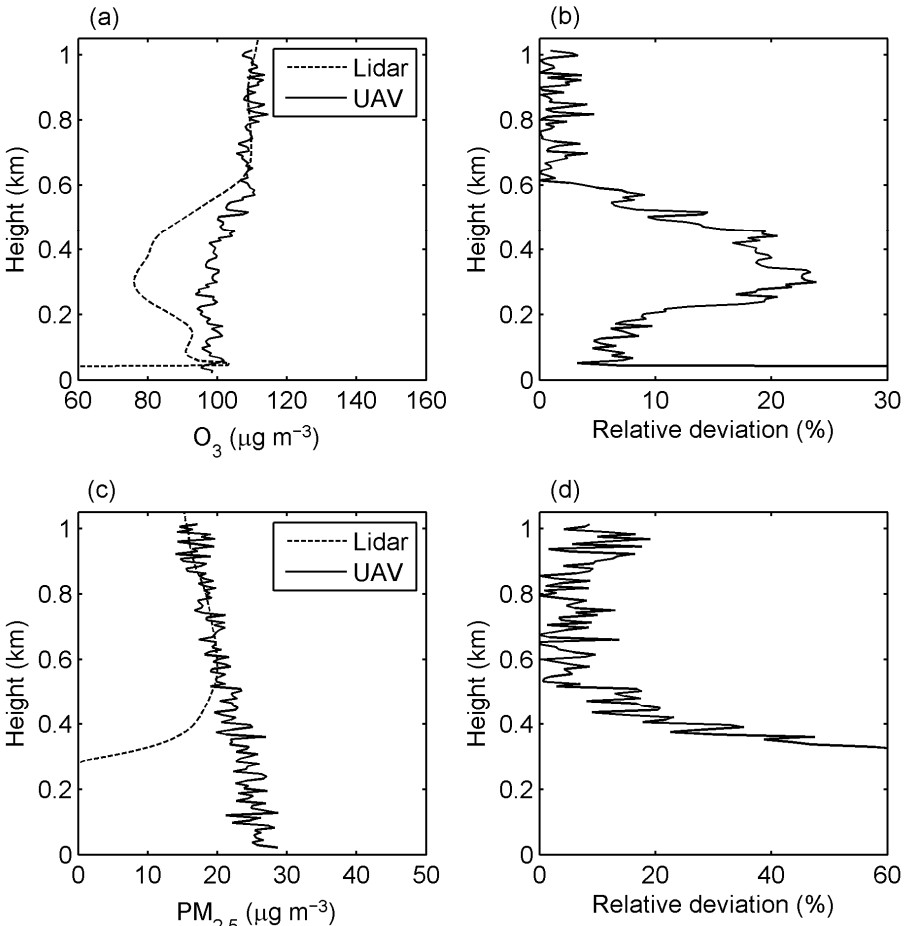

**Figure 2.** Comparison between Lidar and UAV observations. (**a**) $O_3$ profiles obtained by Lidar (dashed line) and UAV (solid line); (**b**) relative deviations of the $O_3$ profiles; (**c**) $PM_{2.5}$ profiles obtained by Lidar (dashed line) and UAV (solid line); (**d**) relative deviations of the $PM_{2.5}$ profiles.

The mobile observation vehicle experiment was carried out from 12 May to 14 May 2019. Figure 3 compares the low-altitude ozone concentration observed by Lidar with the 48 h data series from the mobile observation vehicle. The average ozone concentration at 200 m detected by Lidar (dashed line) varied from 32.4 to 246.6 $\mu g\ m^{-3}$, which is similar to the changing trend of surface ozone concentration monitored by the mobile observation vehicle (solid line), but the value is slightly lower. Similar trends in the two sets of data suggest that Lidar can capture the changing characteristics of ozone concentrations.

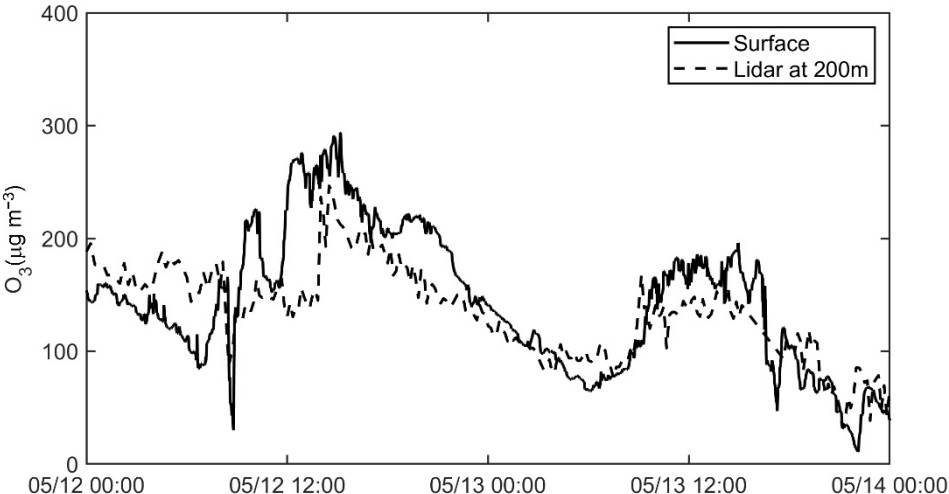

**Figure 3.** Comparison between the surface ozone concentration obtained by mobile observation vehicle (solid line) and Lidar (dashed line).

## 3. Results

### 3.1. Surface Observations

During the observation period from 4 April to 27 May, the average ozone concentration in the eastern suburb of Nanjing was 87.78 μg m$^{-3}$, and the hourly maximum was 274.42 μg m$^{-3}$. The highest daily maximum 8 h average ozone concentration reached 238.76 μg m$^{-3}$, exceeding the Grade-II national ambient air quality standard of 160 μg m$^{-3}$ [26]. The average concentration of PM$_{2.5}$ was 43.48 μg m$^{-3}$, and the hourly maximum reached 155.78 μg m$^{-3}$. The mean diurnal variations in ozone and PM$_{2.5}$ concentrations are presented in Figure 4a,b, respectively. The change curve of ozone concentration showed a single-peak shape and reached the peak at 15:00 LST. The diurnal variation in PM$_{2.5}$ showed a double-peak pattern. The PM$_{2.5}$ concentration reached the maximum during the morning rush hour at around 7:00 LST, dropped to the minimum at noon, and gradually increased after 15:00 LST, reaching the second peak value at midnight.

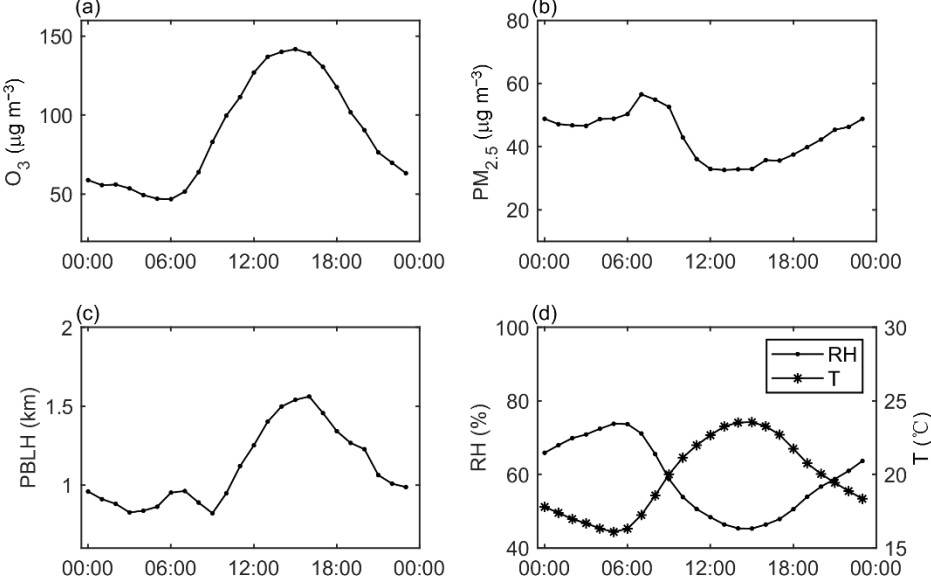

**Figure 4.** The average diurnal variations in (**a**) ozone concentration, (**b**) PM$_{2.5}$ concentration, (**c**) planetary boundary layer height, (**d**) relative humidity and temperature.

We further studied the relationship between surface ozone and particulate matter. Figure 5 shows the correlation between average $PM_{2.5}$ and ozone concentrations during the day (08:00 to 18:00 LST) under different pollution conditions. Figure 5a shows the average $PM_{2.5}$ and ozone concentrations for all 54 days (N = 54) from 4 April to 27 May, and the correlation between $PM_{2.5}$ and ozone is weak, with a correlation coefficient (R) of −0.026. Figure 5b,c present the average pollutant concentrations under the condition that the daytime average $PM_{2.5}$ is less than 40 µg m$^{-3}$ and greater than 50 µg m$^{-3}$, respectively. There are 29 days that meet the condition that the mean $PM_{2.5}$ < 40 µg m$^{-3}$. On these clean days, the ozone concentration significantly positive correlated with $PM_{2.5}$, with correlation coefficient R = 0.51 and $p$ = 0.004. When $PM_{2.5}$ > 50 µg m$^{-3}$, ozone concentration was negatively correlated with $PM_{2.5}$, with R = −0.41 and $p$ = 0.1 (Figure 5c).

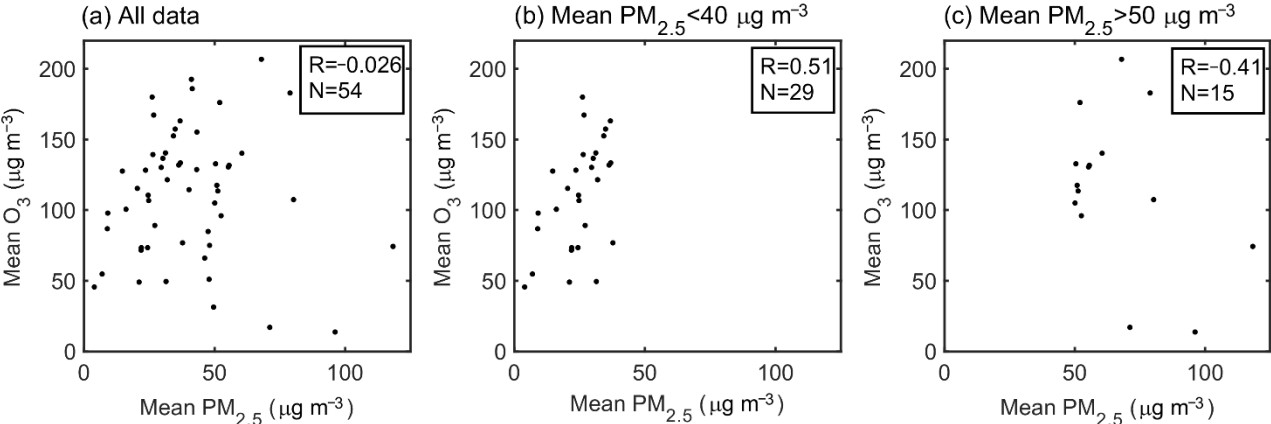

**Figure 5.** The correlation between daytime (08:00 to 18:00 LST) average $PM_{2.5}$ and ozone concentrations for: (**a**) all the 54 days; (**b**) the 29 days that the daytime average $PM_{2.5}$ < 40 µg m$^{-3}$; (**c**) the 15 days that the daytime average $PM_{2.5}$ > 50 µg m$^{-3}$.

*3.2. Lidar Observations*

3.2.1. Vertical Profiles

Figure 6 illustrates the vertical distribution of ozone concentration and aerosol extinction coefficient at 532 nm from 500 m to 3000 m. The average values and the standard deviations at each 37.5 m height were represented by black dots and horizontal bars, respectively. Vertical profiles containing low clouds were not included in this study. Figure 6a shows that the mean ozone concentration was relatively high below 825 m, with a value of 156 µg m$^{-3}$. From 825 m to 1875 m, the ozone concentration decreased from 156.19 µg m$^{-3}$ to 114.85 µg m$^{-3}$ and remained around this value until 2475 m. Between 2475 m and 2775 m, ozone slightly increased by 8.17 µg m$^{-3}$, while the average aerosol extinction coefficient decreased monotonically with height. Particulate matter was mainly distributed in the PBL and with high variability in the concentration. Figure 6b shows that the standard deviation of the extinction coefficient was larger below 1500 m, and decreased rapidly above 1500 m.

The mean diurnal variation in the vertical distribution of ozone concentration is shown in Figure 7a and the curves of ozone concentration variation at different altitudes are shown in Figure S1a. Each curve is the average of the detected ozone concentration within 337.5 m above and below its marked altitude, and the days with precipitation or low clouds were excluded. Within the height of 3000 m, ozone concentration was higher during the day and lower at night, but with the increase in height, the noon peak of ozone concentration became insignificant. Between 700 m and 3000 m, ozone concentration peaked at 12:00 LST, while surface ozone peaked at 162.8 µg m$^{-3}$ at 15:00 LST. The maximum ozone concentration occurred in the upper part of PBL, from 700 m to 1500 m. Figure 7b presents the diurnal variation in the vertical distribution of the aerosol extinction coefficients. The PBLH was calculated by applying wavelet covariance transform to Lidar backscatter profiles and

is represented by the black line in Figure 7b. The PBLH rose in the morning, reached a maximum of 1520 m at 16:00 LST, and dropped to about 750 m at 03:00 LST.

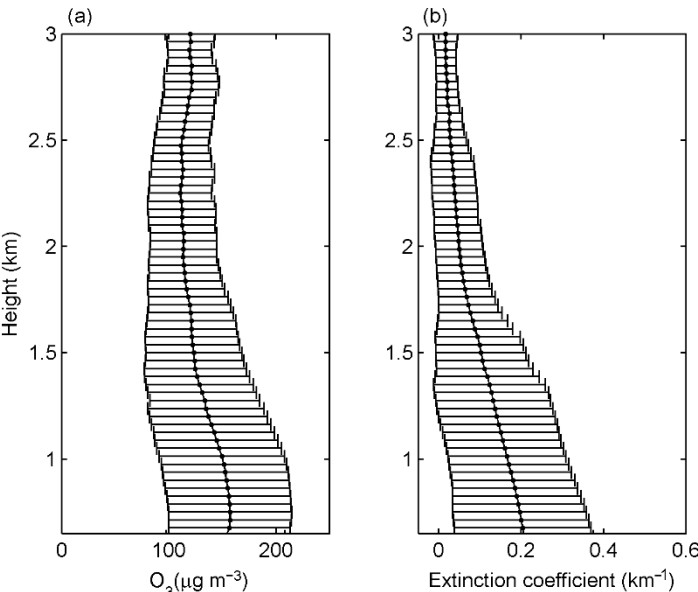

**Figure 6.** Average Lidar profiles of (**a**) ozone and (**b**) aerosol extinction coefficient during the observation period in spring.

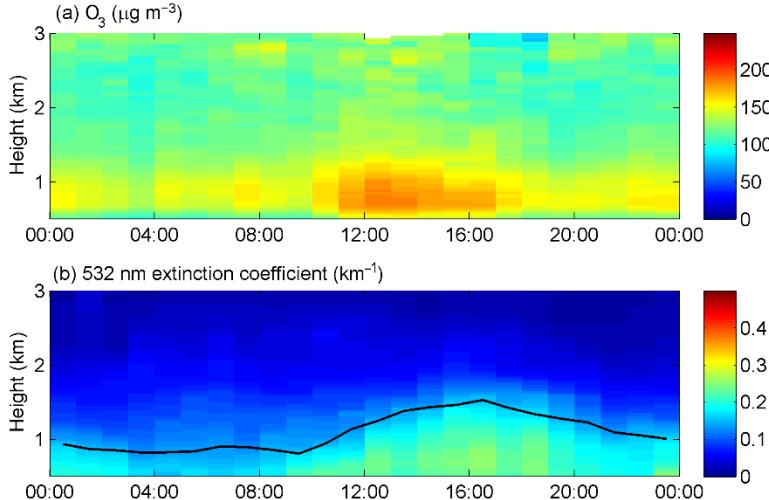

**Figure 7.** Diurnal variations in vertical structures of (**a**) ozone concentration and (**b**) aerosol extinction coefficient during the Lidar observation period in spring. The black line indicates the PBLH.

### 3.2.2. Correlation between Ozone and Particulate Matter

In order to investigate the correlation between ozone and particles, the correlation coefficient of ozone concentrations and aerosol extinction coefficient between 500 m to 3000 m is depicted in Figure 8. We used daily average data in the calculation to eliminate the influence of diurnal variations in solar radiation on ozone and particulate matter. The two pollutants were negatively correlated below 650 m. Within the height of 1300 m, the correlation coefficient between ozone and particulate matter increased with height. At around 1700 m, usually the maximum height of the PBL in the YRD in spring, the correlation coefficient grew considerably, reaching a maximum of 0.379 at 1875 m. Above this height, the correlation coefficient fluctuated but the overall correlation was positive, with an average correlation coefficient of 0.241.

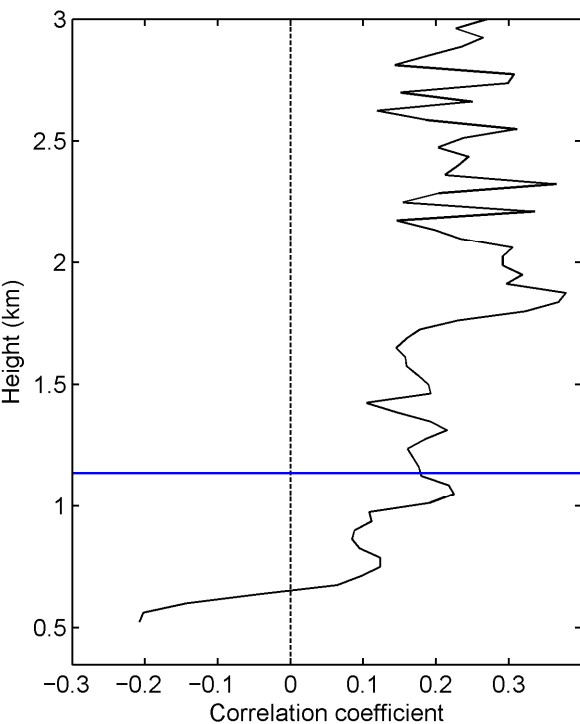

**Figure 8.** The vertical distribution of correlation coefficients between the daily average ozone concentrations and extinction coefficients over the Lidar observation period. The blue solid line represents the average PBLH.

The correlation between the ozone concentration and the aerosol extinction coefficient at different heights under different particulate matter pollution conditions is analyzed in Figure 9. The left panels of Figure 9 show the diurnal variation in the correlation coefficient between ozone concentration and aerosol extinction coefficient, and the right panels present the vertical structure of the aerosol extinction coefficient. Figure 9a,b were calculated based on the data of all observation days, and Figure 9c,d were based on the days with daily average surface $PM_{2.5}$ concentration higher than 35 $\mu$g m$^{-3}$, namely, particulate pollution days. Figure 9e,f were based on the days with average $PM_{2.5}$ concentration lower than 35 $\mu$g m$^{-3}$, namely, clean days. The ozone concentration was positively correlated with the extinction coefficient in the upper layer, with the correlation coefficient about 0.2 to 0.55. They were generally negatively correlated in the lower parts of PBL, and the maximum negative correlation coefficient reaches −0.39. However, there were exceptions to the above findings. A positive correlation between ozone and aerosol extinction coefficient within the PBL could be found from around 11:00 to 16:00 LST, and this positive correlation was more pronounced under low-$PM_{2.5}$ conditions.

A typical particulate matter pollution case was selected in this study to further demonstrate the correlation between ozone and aerosol. The case study period is from 00:00 5 April to 00:00 7 April. Figure 10 shows the vertical characteristics of ozone concentration, aerosol extinction coefficient, and Lidar depolarization ratio. The two-day ozone and aerosol observations showed a "seesaw" pattern, with higher extinction coefficient and lower ozone concentration on the first day, and lower extinction coefficient and higher ozone concentration on the second day. On 5 April, the PBLH gradually rose to 1660 m at around 15:00 LST and fell back after 17:00 LST. The extinction coefficient at 532 nm ranged from 0.21 to 0.38 km$^{-1}$. The ozone concentration reached its maximum of 243.5 $\mu$g m$^{-3}$ under 1500 m at around 12:00 LST. On 6 April, a dust layer with low scattering and high depolarization ratio was observed below 2300 m. From 13:00 LST, the dust layer, with a thickness of more than 1000 m, slowly fell to the ground. Due to the smaller extinction coefficient of the aerosol on the second day and the smaller optical thickness, the shortwave radiation was stronger than that on the first day, which favored the production of ozone.

In addition, the scattering effect of the dust layer could contribute to the photochemical production of ozone. Therefore, the area of high ozone value was expanded in both height and time dimensions compared to the first day. At around 12:00 on 6 April, the ozone concentration at 500 m reached the maximum of 266.9 µg m$^{-3}$.

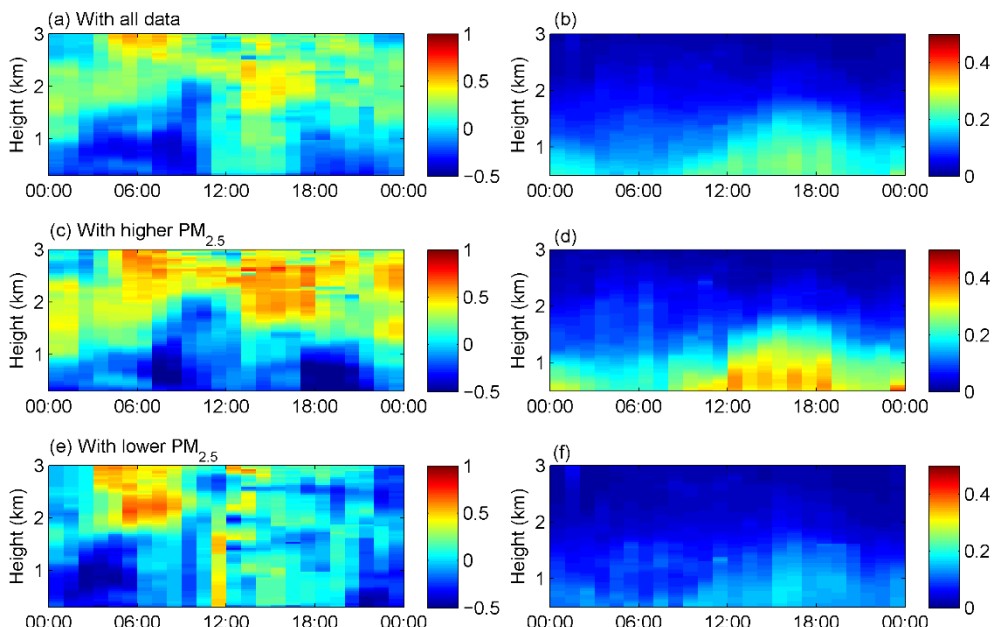

**Figure 9.** Diurnal variations in the correlation coefficient between aerosol extinction coefficient and ozone concentration at different heights (**a**,**c**,**e**) and diurnal variation in aerosol extinction coefficient (**b**,**d**,**f**). The correlation coefficient and aerosol extinction coefficient are calculated from data of all observation days (**a**,**b**), the days with average PM$_{2.5}$ > 35 µg m$^{-3}$ (**c**,**d**), and the days with average PM$_{2.5}$ < 35 µg m$^{-3}$ (**e**,**f**), respectively.

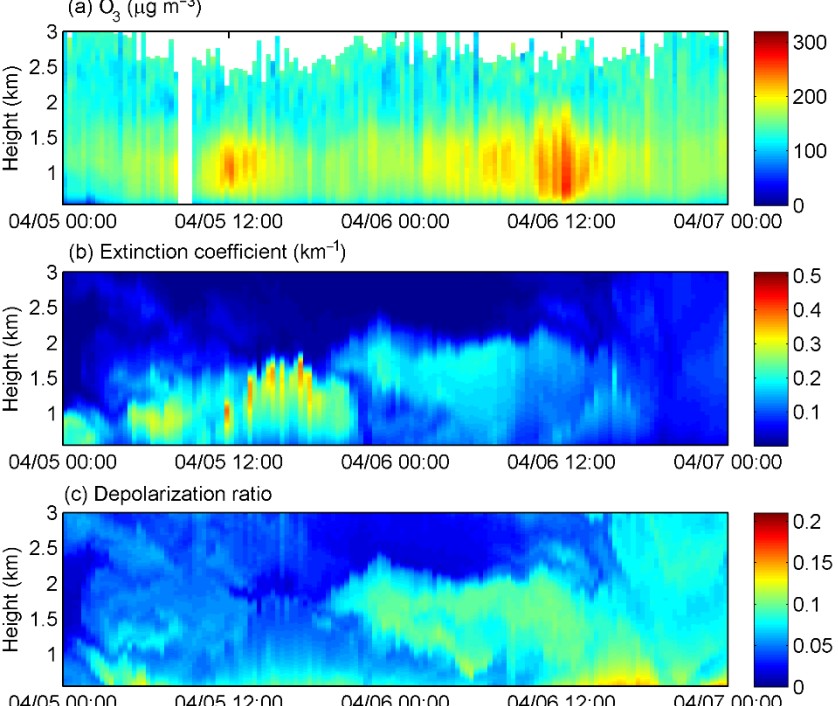

**Figure 10.** Time series of vertical distribution of (**a**) O$_3$ concentration, (**b**) aerosol extinction coefficient at 532 nm, (**c**) depolarization ratio, from 00:00 5 April to 00:00 7 April.

### 3.3. UAV Observations

Due to the geometric overlap factor, Lidar cannot obtain accurate measurements below 500 m. To study the vertical distribution of ozone and particulate matter at lower altitudes, we carried out an UAV experiment on 20 May 2019. The morning was clear and turned cloudy (high clouds above 6000 m) after 13:00 LST. A total of six flights were conducted from 10:00 to 20:10 LST, with each flight taking about half an hour. The maximum flight altitudes of the UAV in the six experiments were 458 m, 518 m, 810 m, 720 m, 1013 m, and 1005 m, respectively. During each flight, the UAV ascended slowly, and the ozone and $PM_{2.5}$ concentration data monitored during the ascent were simultaneously sent back, and then the UAV quickly descended. The data during the descent were not used for this study.

Figure 11 shows the UAV observations and the simultaneous Lidar observations, where the blue line indicates the vertical profile measured by the UAV and the black line indicates the Lidar profile. The upper panel of Figure 11 shows the ozone profile and the lower panel shows the $PM_{2.5}$ profile, and the takeoff time of the UAV is given at the top of each subfigure. For ozone, at 10:00 LST, it reached a maximum concentration of 114 $\mu g \, m^{-3}$ at about 90 m, and decreased slightly from 90 m to 600 m. Due to the residual ozone from the previous day, the ozone concentration above 1200 m was higher than that at low altitudes. Around noon, the ozone in the near-surface layer increased to more than 120 $\mu g \, m^{-3}$, and the trend of decreasing with height extended to about 2000 m. In the afternoon, the near-surface ozone maintained a high value of more than 120 $\mu g \, m^{-3}$. High ozone concentration was observed from 800 m to 1400 m, with a maximum of 129.6 $\mu g \, m^{-3}$ at 1050 m. At 18:01 LST, ozone concentration decreased at all altitudes, especially below 500 m, where ozone depletion was faster due to NO emissions in the evening rush hours. At 19:43 LST, the near-surface ozone further decreased to 73.9 $\mu g \, m^{-3}$, and showed a vertical characteristic of increasing concentration with height.

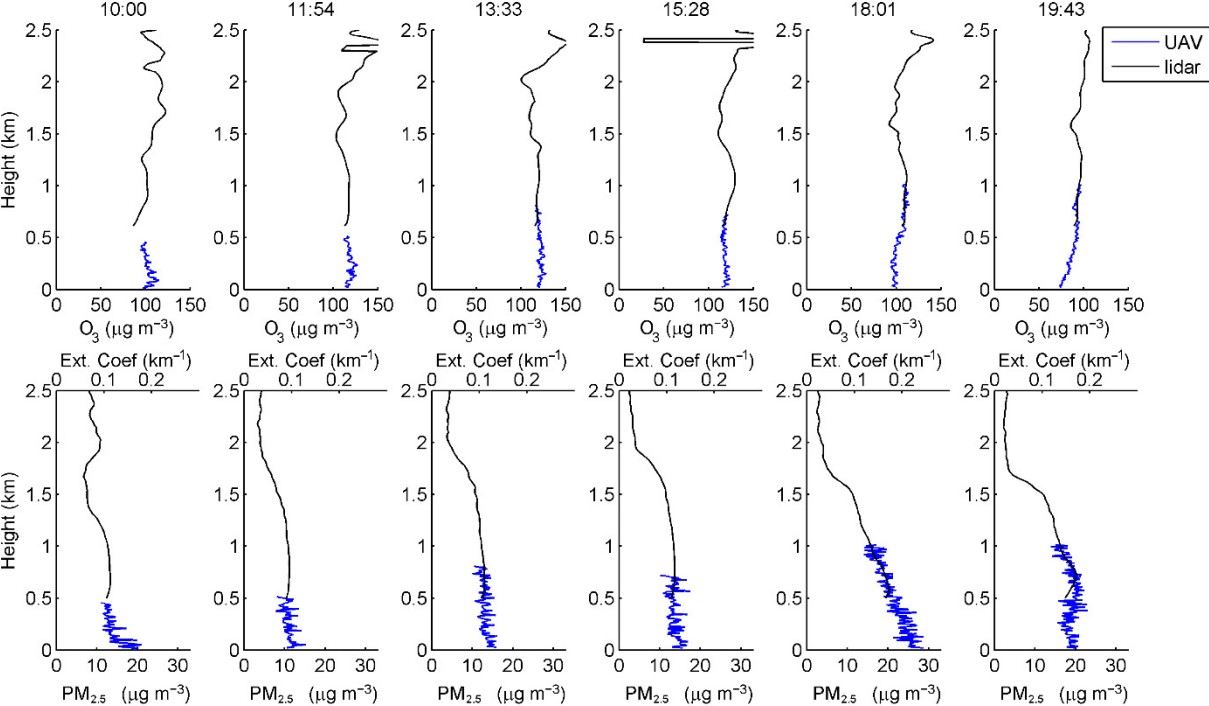

**Figure 11.** Vertical distribution of ozone concentration, $PM_{2.5}$ concentrations, and aerosol extinction coefficient detected by UAV (blue line) and Lidar (black line) on 20 May. The top panel shows the ozone profiles and the bottom panel shows the $PM_{2.5}$ and aerosol extinction coefficient profiles.

Most of the particulate matter was concentrated in the PBL and decreased with height. From 10:00 to 13:33 LST, as the PBLH rose from 1260 m to 1800 m, the near-surface particulate matter was transported upward and well-mixed in the PBL by turbulence,

improving the surface air quality. During the morning, the lowest value of surface $PM_{2.5}$ was about 12 μg m$^{-3}$, which was found at 11:54 LST. After 15:28 LST, particulate matter accumulated in the PBL, and with the weakening of solar radiation, the PBLH gradually decreased to 1650 m by 20:00 LST. The maximum surface $PM_{2.5}$ concentration in the evening was about 28 μg m$^{-3}$, which occurred at 18:01 LST. At 19:43 LST, the $PM_{2.5}$ concentration below 500 m decreased, probably due to the change in wind speed and the change of wind direction in the evening.

## 4. Discussion

Based on mobile observation vehicles, Lidar, and UAV observations, we obtained concentrations of fine particulate matter and ozone in the lower troposphere below 3000 m in the urban area of YRD in spring. This section discusses the distribution characteristics of $PM_{2.5}$ and $O_3$ at the surface and boundary layer, and further discusses the correlation between $PM_{2.5}$ and $O_3$ at different heights and under different pollution conditions within the boundary layer.

### 4.1. Surface

During the observation period, the diurnal variation curve of surface ozone concentration showed a single-peaked shape and the diurnal variation in $PM_{2.5}$ concentration was double-peaked. Diurnal changes in ozone are mainly affected by solar radiation and temperature, therefore both PBLH (Figure 4c) and surface temperature (Figure 4d) showed similar diurnal variation patterns to that of ozone (Figure 4a). Sunrise in Nanjing in spring is around 5:00 LST, when solar radiation begins to rise and heat the ground. As the temperature rises, turbulent mixing develops and the PBLH starts to lift within a few hours. At the same time, since solar radiation is necessary for photochemical reactions, ozone production begins after sunrise and the concentration increases after 6:00 LST. The rise in surface temperature, boundary layer height, and ozone concentration continues until 15:00 or 16:00 LST, and then begins to decline as solar radiation decreases. During the night, the ozone concentration continues to decrease due to the cessation of photochemical production and the influence of $NO_x$ titration [27], and reaches a minimum value before sunrise. Shi et al. [28] also found significant sensitivity of ozone to temperature and PBLH. The diurnal variation in $PM_{2.5}$ is related to PBL. During the daytime, the boundary layer is well-developed, and the movement of atmosphere is favorable to the transport, dispersion, and deposition of pollutants. At night, the low boundary layer and stable atmosphere cause particulate matter to accumulate near the ground and the concentration increases. Figure 4d shows that the fluctuation of relative humidity also highly resembled that of $PM_{2.5}$, and Zhang et al. [29] proved a strong correlation between high humidity and an abrupt rise in $PM_{2.5}$. This could be related to the hygroscopic growth of fine particles and the similar effect of boundary layer atmospheric diffusion conditions on water vapor and particulate matter.

The correlation between surface ozone and particulate matter is not significant. However, when considering different pollution conditions, different correlations exist between particulate matter and ozone. When the $O_3$ concentration is high, and the $PM_{2.5}$ concentration is relatively low (Figure 5b), ozone contributes to the oxidative generation of secondary particulate matter and leads to a positive correlation between the two pollutants. Zawacki et al. [30] found that the emission source of primary particulate matter is similar to that of the ozone precursors $NO_x$ and VOCs, such as mobile source, which may also lead to the positive correlation between ozone and particulate matter. When the $PM_{2.5}$ concentration is high, the attenuation of ultraviolet radiation by particulate matter became a key factor affecting the photochemical reaction, thereby reducing the ozone generation rate, which may lead to a negative correlation between $PM_{2.5}$ and $O_3$ [31].

### 4.2. Boundary Layer

Both Lidar (Figure 6a) and UAV (Figure 11) showed small changes in ozone concentration below 700 m, while from about 700 m to 1500 m, ozone concentration slightly decreased with height. This is mainly due to the fact that, within the boundary layer, the atmosphere is well-mixed, and the difference in the vertical distribution of ozone concentration is small. Above 2500 m, ozone increased due to stronger solar radiation and ozone transport from the stratosphere to the troposphere [32]. At different altitudes, the diurnal variation in ozone concentration is similar to that of the ground during the day but different at night (Figure 7a and Figure S1a). The low nighttime surface ozone concentration was mainly caused by the $NO_x$ titration [27], while at higher altitudes, such as above 700 m, the nighttime ozone concentration remained above 100 $\mu g\ m^{-3}$ due to the lack of NO to react with $O_3$. The maximum ozone concentration occurred in the upper part of PBL, from 700 m to 1500 m, in the afternoon. On the one hand, at this height, due to the lower concentration of particulate matter, the weakening effect of aerosol on solar radiation was weaker than that of the surface. More ultraviolet radiation favored the photochemical production of ozone, so the ozone concentration at this height was higher than that at the surface. On the other hand, since ozone precursors were mainly distributed near the ground, the concentration of ozone precursors at this height was higher than that above the boundary layer, so more ozone could be produced.

Aerosol extinction coefficient decreased with height. Particulate matter was mainly distributed in the PBL and with high variability in the concentration. Figure 6b shows that the standard deviation of the extinction coefficient was larger below 1500 m, and decreased rapidly above 1500 m, which is similar to the PBLH found by Huo et al. [33] in spring in eastern China. During the daytime, air pollutants within the PBL are well mixed by convection induced by the surface heating. Particulate matter near the surface can be transported upward by turbulence, which increases the particulate matter concentration in the upper boundary layer and improves air quality near the surface [34]. Therefore, the variation trend of the aerosol extinction coefficient in the upper 700 m to 1500 m of the PBL (Figure S1b) was similar to that of the PBLH (Figure 7b) and opposite to that of the surface $PM_{2.5}$ concentration (Figure 4b). Since the particulate matter was largely allocated within the PBL, the aerosol extinction coefficient was small above the PBL and remained stable throughout the day.

The correlation between aerosols and ozone varies at different heights (Figure 8). The negative correlation below 650 m was probably owing to the particle-induced reduction in shortwave radiation. When shortwave radiation is reduced, the photolysis rate drops, resulting in less photochemical production of ozone [35]. With the increase in height, especially above the aerosol layer, on the one hand, the attenuating effect of particulate matter on shortwave radiation and ozone generation is weak. On the other hand, the backscattering of particulate matter increases the photolysis rate in the upper layer and facilitates ozone production [36]. The correlation is also different under different pollution conditions (Figure 9). When there is more particulate matter in the PBL, the weakening effect of particulate matter on the downward shortwave radiation is stronger, and the photolysis rate in the boundary layer decreases more, which is less conducive to the generation of ozone. Therefore, the negative correlation between ozone and aerosol extinction coefficient is stronger in the case of high $PM_{2.5}$ than in the case of low $PM_{2.5}$. Correspondingly, above the boundary layer, the scattering effect of aerosol is stronger when there are more particles, and the increased backscattered shortwave radiation is beneficial to the photochemical production of ozone. Thus, the positive correlation coefficient above PBL is higher when the particulate pollution is heavier. A positive correlation can also be found in the PBL in the afternoon. At this time, the solar radiation is strong and the secondary aerosol formation induced by rich ozone plays a dominant role, while the scattering or extinction effect of aerosol is less competitive.

## 5. Conclusions

In order to study the vertical distribution of ozone and particulate matter in the lower troposphere in spring, we conducted in situ observations in Nanjing, China, between 4 April and 27 May. Surface data were obtained from a parked mobile observation vehicle, and vertical data were collected from continuous observation by Lidar. The Lidar measurements were validated by comparing with the pollutant profiles obtained from the UAV observations and the surface observations.

During the observation period, the average surface ozone and $PM_{2.5}$ concentrations were 87.78 $\mu g\,m^{-3}$ and 43.48 $\mu g\,m^{-3}$, respectively. Surface ozone and $PM_{2.5}$ were positively correlated with a correlation coefficient of 0.51 when $PM_{2.5} < 40\ \mu g\,m^{-3}$ and negatively correlated with a correlation coefficient of $-0.41$ when $PM_{2.5} > 50\ \mu g\,m^{-3}$.

The Lidar observation shows that within 3000 m, the highest ozone concentration can be found in the upper PBL. Particulate matter is mainly distributed in the PBL and decreases monotonically with height. Ozone and aerosols show diurnal variation at all altitudes below 3000 m, but the variation is not significant above the PBL. The correlation between ozone and aerosol is different at different altitudes and under different particle pollution conditions. Below 650 m, the daily average ozone concentration was negatively correlated with the aerosol extinction coefficient, and the correlation coefficient increased with altitude, reaching a maximum value of 0.379 at 1875 m. Within the PBL, on the particle pollution days (daily average $PM_{2.5}$ more than 35 $\mu g\,m^{-3}$), ozone and aerosols are negatively correlated, while on the clean days they are positively correlated. Above the PBL, the correlation coefficient is usually positive regardless of whether it is a particulate pollution day. The correlation coefficient reflects the competitive relationship between the increasing and inhibiting effects of particulate matter on ozone production.

The UAV case study on 20 May compensated for the overlap area of Lidar detection within 500 m. The UAV matched the Lidar results within the altitude that could be covered by both observation methods. By combining the UAV and Lidar observations, it was found that the ozone concentration in the upper layers was higher than that in the near-surface layers, and that the ozone depletion in the near-surface layers was faster than that in the upper layers after sunset.

In this study, the vertical distribution of ozone and particulate matter in the lower troposphere of Nanjing in spring and the correlation between them were investigated. Particulate matter has a complex effect on ozone. The extinction and scattering effects lead to a decrease in ozone concentration in the lower boundary layer and an increase in ozone concentration in the upper boundary layer, so that ozone is negatively correlated with particulate matter in the lower layer and positively correlated in the upper layer. This study mainly focused on spring, and in the future, more research should be carried out in the summer, with high ozone and low particulate matter concentration, and in the winter with low ozone and high particulate matter concentration. In addition, more methods, such as satellite observations and atmospheric sounding, can be used to improve the study of the vertical distribution and correlation of particulate matter and ozone, and model simulations should be used in future studies to explore the underlying mechanisms.

**Supplementary Materials:** The following supporting information can be downloaded at: https://www.mdpi.com/article/10.3390/rs14133051/s1, Figure S1: Diurnal variations in (a) ozone concentration and (b) aerosol extinction coefficient at different height during the Lidar observation period in spring.

**Author Contributions:** Conceptualization, T.W.; methodology, Y.Q. and M.Z.; data curation, S.L.; validation, M.L., M.X. and B.Z.; writing, Y.Q.; visualization, M.Z.; supervision, T.W.; funding acquisition, T.W. and Y.Q. All authors have read and agreed to the published version of the manuscript.

**Funding:** This research was funded by the National Natural Science Foundation of China (42077192, 41621005), the Joint Open Project of Key Laboratory of Meteorological Disaster, Ministry of Education & Collaborative Innovation Center on Forecast and Evaluation of Meteorological Disasters, NUIST

(KLME202109), the National Key Basic Research & Development Program of China (2020YFA0607802, 2019YFC0214603), and the Talent Introduction Project of Jinling Institute of Technology (No. jit-b-202108).

**Data Availability Statement:** The data presented in this study are available on request from the author.

**Acknowledgments:** We would like to thank the Emory University–Nanjing University Collaborative Research Grant for providing valuable support to this study.

**Conflicts of Interest:** The authors declare no conflict of interest.

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
