# Peer review of "Lidar- and UAV-Based Vertical Observation of Spring Ozone and Particulate Matter in Nanjing, China"

_remotesensing, doi:10.3390/rs14133051_

Round 1
Reviewer 1 Report
Review of Manuscript: remotesensing-1773730
Lidar and UAV-based vertical observation of spring ozone and particulate matter in Nanjing, China; by Y. Qu et al.
Highlights
- Measurements of PM and O3 using Lidar and UAV, with their vertical information.
- Analysis of vertical and diurnal change characteristics between PM and O3.
Concerns
- Uncertainty of measurement may be still large.
- The data support the conclusion is weak and may lack statistical significance.
General comments
The manuscript study vertical and diurnal change of PM and O3 using Lidar and UAV observations.
The purpose of the research is to understand potential relationship or interaction between pollutant PM and O3, in a perspective of statistics. Weak correlations between them are found, and they are negative near surface and positive at relatively high attitude. This complements the co-variation characteristics of regional air pollutants. The specific comments are showing as follows:
Specific comments
1) Line 23: please remove word ‘monotonically’.
2) Line 42: I agree that the increasing surface ozone has become a severe environmental problem in current. While, can I think that the decrease in PM will promote the increase in O3? as you presented in Abstract (Line 25-26). How to define a day with particulate pollution?
3) In the third paragraph of the introduction, I encourage the authors to summarize and generalize these research findings, rather than describe them one by one. Here is another research about vertical particle that may help (10.5194/acp-21-17003-2021).
4) Line 106: ‘April 4 to May 27, 2019?’; year number is missing.
5) Line 158: What is the initial spatial resolution of UAV? which interpolated method is choose?
6) Figures are not enough clear and it make difficult for me to read the manuscript.
7) Is there any comparison between your measurements and the China National Environmental Monitoring Center (CNEMC)?
8) Which instrument does the data come from in Figure 4 (Lidar, UAV, or average) and what is the limitation of height for the ‘surface’? Please clarify. Because if the gravimetric method is used for PM, the particles are dried before being measured and thus this explanation (Line 203) doesn't make sense.
9) Line 216, R=0.51 is a weak correlation. Please clarify if the ‘significantly positive’ here means that all regressions (is linear?) are tested for significance. I also recommend authors perform a significance test for all weakly correlated regression. For only my perspective, authors also can use log(x) instead of x, if linear regression is not well.
10) The PM in Figure 4 is opposite to the 532 nm extinction coefficient in Figure 7 in diurnal.
11) Only Lidar measurement are used in Figure 8? What about UAV measurement.
12) Line 286-289; I am not sure to agree or understand this. If the R is calculated by daily average data during the study period, how to obtain diurnal features including incoming solar radiation, since time information is averaged in a day.
13) Line 291: I recommend show the average PBLH in Figure 8 for better reading.
14) I am appreciating to the Figure 9. In the diurnal characteristics, from your previous explanation, is the incoming solar radiation a more important factor? Especially in cloudy days.
15) Section 3.3 should go where 3.2.1, as they are both about vertical features of PM and O3. Or add a subsection on relationship between PM and O3 using UAV data at the end of Section 3.
Author Response
We would like to thank the reviewer for the valuable comments. We have made changes to the relevant figures and descriptions as suggested and provided point-by-point responses. Please see the attachment.

Reviewer 2 Report
This is an interesting paper which studied vertical distribution of ozone (O3) and fine particulate matter (PM2.5) by lidar and UAV-based observations in spring in Nanjing and discussed their interactions in order to control both air pollutions synergistically. The authors found moderate concentrations of surface O3 (88 μg m-3) and PM2.5 (43 μg m-3) as well as some trade-offs of the two air pollutants in temporal and spatial scales in their study period. The combination of the two methods: UAV and lidar provided a new understanding for both air pollutants because the UAV case study could compensate for lidar detections below 500 m. Another important finding was that O3 concentration was higher in the upper layers than in the near-surface layers, which means O3 depletion was faster in the near-surface layers especially after sunset. I believe this study fits well the scope of Remote Sensing (RS) and merits to be published in the RS journal after minor revisions (see below).
L377-378: Replace “At 19:43-3,” by “At 19:43 LST,”
Figs. 6, 7, 10 and 11: unit of O3 concentration should be corrected to “μg m-3” (-3, superscript)
Fig. 11: PM2.5 in X-axis (2.5, subscript); concentration of PM2.5 to be “μg m-3” (-3, superscript)
Author Response
We sincerely appreciate the reviewer for the valuable comments. In the revised manuscript, we have corrected the typos and units in the figures as suggested. We have also made some other changes to the manuscript. Please do not hesitate to let us know if you have additional comments. Thanks again.
Reviewer 3 Report
First of all, the observation data obtained in the study is very valuable, and the topic is also very interesting. The rapid urbanization in China is accompanied by increasingly serious air pollution. Particulate matter and ozone are the main air pollutants, and the study of their vertical distribution and correlation plays an important role in the synergistic air pollution control. The paper is well structured and the results clear and well discussed. Therefore, also given the importance of the topic dealt with, I recommend accepting the paper for publication after making the modifications listed below.
1 Line 169, “The ozone profiles from the two methods were in good agreement from 500 m to 1000 m, with a relative deviation of less than 5%”, and “The PM2.5 concentration profiles observed above 500 m by UAV and Lidar were similar, with a relative deviation of less than 20%.” I think there is a huge difference between 5% and 20%, please clarify the deviation of PM2.5 concentration.
2 Line 182, The solid line and dotted line in Figure 3 are not clearly distinguished.
3 Lines 194-197, figure 2b presents that the second maximum value is not at midnight, peak value and maximum value is different, please redescribe the PM2.5 concentration.
4 Lines 197-204, The diurnal variation of ozone and PM2.5 concentration are not well explained.
5 Line 227, Figure 5 is not standardized, for example, the legends in the previous figures are framed, but not here. The writing of PM2.5 is not standardized, etc.
6.Lines 208-225, I think there are too few samples to support the conclusion. In addition, “there are only 12 samples under the polluted condition”, in fact, there are 15 samples in the figure 5c.
7.Lines249-279, in the analysis of diurnal variation of the vertical distribution of ozone and PM2.5 concentration, many descriptions are superfluous. For example, ozone and PM basically exist in the boundary layer, it is little significance to explain the characteristics of 3000 m.
Author Response
We would like to thank the reviewer for the comments and suggestions. We have revised the manuscript accordingly and provided a point-by-point response. Please see the attachment.

Reviewer 4 Report
The paper is easy to read and written in a quite clear mode. It deserves to be published.
Author Response
We would like to thank the reviewer for the review and feedback. We have made some changes to the manuscript. Please do not hesitate to let us know if you have any other questions. Thanks again.
Reviewer 5 Report
This study investigated the vertical distribution of ozone and particulate matter in the lower troposphere of Nanjing in the spring and their association.
The authors found that within the boundary layer ozone and aerosols have a negative correlation on days with particulate pollution, while on clean days they are positively correlated. Above the boundary layer, the correlation coefficient is usually positive, regardless of the presence of particle contamination. The authors state that the ozone concentration is higher in the upper layers than in the near-surface layers, and that ozone damage is faster in the near-surface layers after sunset.
This study focused mainly on spring, and the authors suggest that future research should be conducted in the summer with high ozone and low particulate concentrations and in the winter with low ozone and high particulate concentrations.
The scientific topic discussed in this article is a current and relevant academic topic and explores some of the most important current bibliography in the area. The structure of the article is appropriate and linking the results to the text. Discussion is conveniently linked to the object that has been proposed as a research question, in order to apply the theory to a consistent work of scientific research. Overall, the paper is acceptable.
Author Response
We would like to thank the reviewer for the valuable feedback. We have made some changes to the manuscript. If you have any additional comments, please do not hesitate to let us know. Thanks again.
Round 2
Reviewer 1 Report
I am grateful to the author for replying all my comments and revising the manuscript. Now I think this study is suitable for publication in the journal of RS.